# Efficient Genome Editing in *Bacillus licheniformis* Mediated by a Conditional CRISPR/Cas9 System

**DOI:** 10.3390/microorganisms8050754

**Published:** 2020-05-17

**Authors:** Youran Li, Hanrong Wang, Liang Zhang, Zhongyang Ding, Sha Xu, Zhenghua Gu, Guiyang Shi

**Affiliations:** 1Key Laboratory of Industrial Biotechnology, Ministry of Education, School of Biotechnology, Jiangnan University, Wuxi 214122, China; liyouran@jiangnan.edu.cn (Y.L.); wanghanrong@163.com (H.W.); zhangl@jiangnan.edu.cn (L.Z.); zyding@jiangnan.edu.cn (Z.D.); xusha1984@jiangnan.edu.cn (S.X.); guzhenghua2011@163.com (Z.G.); 2National Engineering Laboratory for Cereal Fermentation Technology, Jiangnan University, Wuxi 214122, China; 3Jiangsu Provincial Research Center for Bioactive Product Processing Technology, Jiangnan University, Wuxi 214122, China

**Keywords:** *Bacillus licheniformis*, CRISPR/Cas9, conditional genome editing, xylose operon

## Abstract

*Bacillus licheniformis* is widely used to produce multiple enzymes and chemicals in industrial fermentation. It is also an organism that is hard to genetically manipulate, which is mainly attributed to its extremely low transformation efficiency. The lack of genetic modification technology severely limits its further application. In this study, an all-in-one conditional clustered regularly interspaced short palindromic repeat (CRISPR)/Cas9 plasmid was developed for *B. licheniformis* with the *cas9* gene under the control of a xylose-inducible promoter. By means of this design, the expression of the *cas9* gene could be repressed without xylose, which significantly improved the transformation ratio from less than 0.1 cfu/μg to 2.42 cfu/μg DNA. Compared with this conditional system, a constitutive overexpression system led to significant growth retardation in bacterial cells. Both the biomass and specific growth rate decreased greatly. After transformation, successful genome editing could be triggered by 0.5% xylose. When the α-amylase gene *amyL* was used as a genomic target, the efficiencies of its disruption using three different protospacer-adjacent motif (PAM) sequences were 64.3%, 70.9%, and 47.1%, respectively. Moreover, temperature plays a pivotal role in the function of the constructed CRISPR system. The maximum success rate reached 97% at 20 °C, while higher temperatures negatively impacted the function of the system. These results suggested that the design with a *cas9* gene under the strict control of a xylose-inducible promoter significantly improved the success rate of genome editing in this host. This work contributes to the development of genetic manipulation and furthers the use of *B. licheniformis* as an efficient industrial workhorse.

## 1. Introduction

*Bacillus licheniformis*, which is a gram-positive bacterium of the phylum Firmicutes, plays a versatile role in the fermentation industry. This strain is equipped with abundant enzyme systems, has a strong secretory ability, has simple cultivation requirements, and, most importantly, has been designated as a generally recognized safe (GRAS) organism for use in the food industry. *B. licheniformis* and its closely related species *Bacillus subtilis* together contribute to nearly half of the global enzyme industry output [1,2]. This bacterium also produces a wide array of valuable metabolites such as peptides, lipopeptides, phospholipids, amino acids, and nucleic acids [3]. However, its ability to produce foreign-gene encoded products is weak, especially compared with that of *Escherichia coli*.

The extensive application of *E. coli* in both research and industrial fields is attributed to the simplicity and effectiveness of the genetic manipulation of this species. Unfortunately, the effective introduction of exogenous DNA or the manipulation of genomic DNA in *B. licheniformis* has proven to be highly challenging [4,5,6]. This can be attributed to multiple reasons, including the lack of natural competence, unique membrane structure, and a robust DNA restriction system [5,7,8]. The currently used main methods for genome editing in *B. licheniformis*-related species are based on plasmids that are temperature sensitive for replication [9,10]. Successful editing requires two-step recombination events and, due to low efficiency in the recovery of antibiotic markers, screening for positive transformants is a time-consuming and cumbersome process. Therefore, new methods for genome editing in this host are urgently needed.

The clustered regularly interspaced short palindromic repeat (CRISPR) system has been identified as a widespread immune system component in a wide variety of species [11,12,13]. The CRISPR-associated (Cas) proteins, in addition to CRISPR RNA (crRNA) and transactivating CRISPR RNA (tracrRNA), can recognize and introduce double-strand breaks (DSBs) in foreign nucleic acids with the aid of a protospacer-adjacent motif (PAM) sequence, which is the spacer region in the single guide RNA (sgRNA) that is highly specific for the targeted gene [14,15]. This system is functional in both prokaryotes and eukaryotes with the difference being that a donor fragment is mandatory when performing genomic mutations in prokaryotes because most prokaryotic species lack the nonhomologous DNA end-joining (NHEJ) recombination system that is, in contrast, universal in eukaryotes [16]. Considering the above factors, genome editing would leave only the genotype that harbors the designed donor fragment. A type II CRISPR/Cas9 system from *Streptococcus pyogenes* was chosen and redesigned to integrate a chimeric sgRNA (containing features of both crRNA and tracrRNA) with an artificial donor template (to remedy the DSB caused by a Cas protein) in one plasmid. Based on the above design, the CRISPR/Cas9 system has been widely engineered for either gene deletion or insertion in multiple bacteria [17,18,19]. However, in all cases, high transformation and recombination efficiencies are necessary for successful genome editing. Unfortunately, the low transformation efficiency of *B. licheniformis* makes it almost impossible to achieve one-step genome editing with the CRISPR/Cas9 system. To date, the CRISPR/Cas9 system has been applied for genome editing purposes in very few bacterial species in which transformation or recombination is challenging [9]. Vector size and expression of the harbored gene are two main factors affecting the transformation rate. The former determines the success of transmembrane transport, while the latter exerts pressure on cell propagation [20]. Recently, Li et al. integrated a CRISPR-Cas9 nickase into the genome of *B. licheniformis DW2*, which successfully preserved the size of the genome-editing vector to guarantee a certain transformation rate [10]. Although overexpression of the integrated nickase gene driven by the strong constitutive promoter P43 ensured successful genome editing, it also caused growth retardation in the recombinant strain when compared with the wild-type strain. Moreover, estimating the effect of a foreign gene’s sustained overexpression on the overall physiological metabolism of bacterial cells is difficult. Alternatively, the pattern of inducible expression can minimize the influence of foreign-gene expression on host cell propagation or growth. To date, the strictness of the xylose-inducible expression system has been greater than that of other inducible expression systems, such as *T7*, *spac*, and *grac* at the transcriptional level in *B. licheniformis* [1]. In this study, the type II CRISPR/Cas9 system from *S. pyogenes* was incorporated into a thermosensitive shuttle vector to enable the genome editing of *B. licheniformis*. The *cas9* gene, encoding a DNA endonuclease, was under the strict control of a *B. licheniformis* xylose-inducible promoter, which significantly improved the success rate of genome editing in this host.

## 2. Materials and Methods

### 2.1. Bacterial Strains and Growth Conditions

The bacterial strains and plasmids used in this study are listed in Table 1. The reagents and medium for *Bacillus* transformation were prepared according to Waschkau. *E. coli* and *Bacillus* were grown in errific broth (TB) based on Li’s methods [1]. Then, 250-mL shaking flasks with baffles containing 25 mL of medium were inoculated with specific strains. Then, 100 μg/mL ampicillin was added when necessary to maintain the plasmids in *E. coli*. *Bacillus* transformants were grown with 10 μg/mL erythromycin or 20 μg/mL tetracycline. Cultivation was performed at 37 °C in Luria-Bertani (LB) broth unless otherwise stated.

### 2.2. Reagents

Restriction enzymes and T4 DNA ligase were purchased from Thermo Fisher (Waltham, MA, USA). PrimeSTAR HS and Ex Taq DNA polymerases were obtained from TaKaRa (Dalian, China). Kits for plasmid extraction, gel extraction, and nucleic acid purification (AxyPrep Series) were purchased from Axygen (Hangzhou, China). The Gibson assembly kit was purchased from NEB (Ipswich, MA, USA). All other reagents used in this study were obtained from Sinopharm (Shanghai, China).

### 2.3. Recombinant Vector Construction

The gram-positive plasmid pNZT1 is a replication-thermosensitive derivative of the rolling-circle plasmid pE194. For sgRNA expression, cassettes including a *P_HpaII_* promoter from pMA5 and a 20-bp region complementary to the target region were synthesized as separate fragments by GENEWIZ (Suzhou, China). The oligonucleotides and primers used in this study are listed in Table 2. The cassettes were then cloned into pNZT1 by Gibson assembly using the primer pairs pNZT-F/R and sgRNA-F/R to generate the pN-sgRNA1/pN-sgRNA2/pN-sgRNA3 plasmids. The target loci of sgRNA1-3 correspond to the regions in PAM621 (621 bp downstream of the ATG initiation codon), PAM832 and PAM984, respectively, of the α-amylase gene *amyL* (Gene ID: 3031010).

For *cas9* gene expression, the xylose isomerase regulon *xylA*, which contains the P_xyl_ promoter and *xylR* repressor (locus tag: BL03942) from *B. licheniformis* DSM13, was chosen as the controllable expression element. *xylA* was cloned from genomic DNA using the primer pair xylA-F/R and PrimeSTAR HS DNA polymerase, and the Cas9 protein-encoding gene was amplified from the pCas9 vector using the Cas9-F/R primers and the same enzyme. These two fragments were then successively inserted into the cloning vector pMD19-T at the *Sal*I restriction site by Gibson assembly, where the *cas9* gene was located downstream of *xylA* for conditional expression to yield pM-xylACas9. Then, this plasmid was digested by *Sal*I, and the fragment conferring the *cas9* gene expression cassette was electrophoretically separated and recovered. The cassette was cloned into the pN-sgRNA1, pN-sgRNA2, and pN-sgRNA3 plasmids linearized with *Sal*I to obtain the constructs pN-sgRNA1-xylACas9, pN-sgRNA2-xylACas9, and pN-sgRNA3-xylACas9, respectively.

A donor fragment was designed to repair DSBs. For this purpose, two fragments corresponding to the 5′-ends and 3′-ends of the target gene were cloned using the homology-arm primer pairs HA1-F/R and HA2-F/R. These fragments were assembled by overlap PCR with the assistance of the primer pair HASOE-F/R to yield a truncated version of *amyL* and *truncamyL*. This fragment was then inserted into the plasmid pN-sgRNA-xylACas9 at the *Xho*I restriction site to complete the construction of the genome-editing vectors pN-xylamyLD100, pN-xylamyLD101, and pN-xylamyLD102.

### 2.4. Transformation of the All-in-One Conditional CRISPR/Cas9 Plasmid and Conditional Genome Editing

The complete CRISPR/Cas9 system was constructed in a single plasmid to be used for genome editing in *B. licheniformis* with only one transformation process, as illustrated in Figure 1. Each of the three amylase-knockout plasmids (pN-xylamyLD100, pN-xylamyLD101, and pN-xylamyLD102) contained a *cas9* gene controlled by a xylose-inducible promoter, an sgRNA controlled by the gram-positive promoter *P_HpaII_*, and a donor DNA fragment for DSB repair. Therefore, the synthesis of the 158-kDa Cas9 protein could remain inactivated during the construction and transformation of the plasmids. Afterward, the addition of xylose to the medium initiated synthesis during genome editing. The *P_HpaII_* promoter is a constitutive promoter that strongly drives the transcription of sgRNA in *B. licheniformis*. The DNA donor fragment was obtained by using an overlap PCR of the regions upstream and downstream of the homologous arms, targeting the 71-460 bp and 1101-1478 bp loci, respectively. The assembled *truncamyL* fragment was introduced into the *Xho*I site at the target locus.

We previously constructed a genome-editing plasmid, pN-43amyLD100, with *cas9* under the control of a constitutive promoter, P43, targeting the PAM621 locus of *amyL*. This plasmid and those obtained in this study were introduced into *B. licheniformis* DSM13 by protoplast transformation, according to Li’s method [1]. Positive transformants were screened on a DM3 agar plate with 10 μg/mL erythromycin, which obtained BLG110 (targeting PAM621), BLG107 (targeting PAM621), BLG108 (targeting PAM832), and BLG109 (targeting PAM984).

For conditional genome editing in BLG107, BLG108, and BLG109, the transformants were inoculated into TB medium supplemented with 10 μg/mL erythromycin and cultured at 30 °C with continuous shaking at 200 rpm. When the cells entered the late-log phase, xylose was added at a concentration of 0.5% to activate the expression of the *cas9* gene. Cultivation was continued for 24 h, and an aliquot of cells was then transferred into fresh TB medium for another round of cultivation, with the only difference being that xylose was added at the beginning of cultivation. After 24 h, the cell culture was plated and propagated on TB medium with 10 μg/mL erythromycin at 30 °C. The positive mutants were confirmed using diagnostic PCR with the primers diag-F/R.

To cure the positive mutants of the thermosensitive genome-editing plasmids, the colonies were first cultured in TB medium containing 10 μg/mL erythromycin at 30 °C. After 24 h, the cell culture was inoculated into fresh medium and kept at 42 °C with aeration for another 24 h. Cells were then plated on TB medium with no antibiotics and cultured at 42 °C. Since the erythromycin selection marker was delivered by only the plasmids, the above plate was then printed to TB medium containing 10 μg/mL erythromycin. Therefore, the mutants sensitive to the antibiotic could be identified as those lacking the plasmids.

### 2.5. Biomass and Amylase Assay

The constructed recombinant *B. licheniformis* strains were cultured overnight, and fermentation flasks were then inoculated with 1% of the culture and incubated at 37 °C by shaking at 250 rpm. Samples were collected for OD600 measurements and enzyme assays after fermentation. Biomass was monitored by measuring the absorbance at 600 nm (OD600) with an Ultrospec 3000 spectrophotometer (Pharmacia Biotech, Piscataway, NJ, USA). The cell culture samples were diluted to suitable concentrations before detection to maintain the absorption reading range of the spectrophotometer between 0.2 and 1.2. The α-amylase activity in the culture broth was assayed based on Zhang’s method [23].

### 2.6. Statistical Analysis

All experiments were performed at least three times, and the results are expressed as the means ± standard deviations (SDs). Statistical analyses were performed using Student’s t-test and ANOVA followed by Tukey’s post hoc test. *p* values less than 0.05 indicated significant results.

## 3. Results

### 3.1. Inactivation of cas9 Transcription Contributes to the High Transformation Rates of the Genome-Editing Plasmids

When the *cas9* gene was expressed during the process, with either a constitutive pattern (*P43*) or xylose induction, the transformation rate was clearly much lower than the control’s transformation rate (pNZT1) and achieved with a silent *cas9* design (Figure 2A) despite the plasmids being similar in size. The genome-editing plasmids were transformed into *B. licheniformis* using protoplast transformation, which can be applied to a small strip of cells with high efficiency and minimal cell death. Three plasmids with a conditional design, namely, pN-xylamyLD100, pN-xylamyLD101, and pN-xylamyLD102, achieved a transformation rate between 2.21 and 2.42 cfu/μg DNA. In contrast, pN-43amyLD100, which is a plasmid with a constitutive design, yielded a transformation rate < 0.1 cfu/μg DNA after several failed attempts (Figure 2A). Similarly, when the transformation of plasmid pN-xylamyLD100-102 was carried out under induced conditions, a sharp decline in the transformation rate was observed. All three genome-editing plasmids were associated with great difficulty in acquiring the transformants (<0.1 cfu/μg DNA), while the control (pNZT1) plasmid still exhibited a much higher transformation rate of 2.61 cfu/μg under the same induction conditions. Considering that the sizes of the genome-editing plasmids were larger than that of pNZT1 (~10,000 bp versus 4069 bp), the expression of heterologous genes appeared to have a stronger impact on the transformation rate than the plasmid size in this case. The inactivation of *cas9* transcription clearly contributed to increased transformation rates of the genome-editing plasmids.

The effects of constitutive/conditional designs on cell growth were further compared using the recombinant strains BLG100, BLG107, BLG108, BLG109, and BLG110 harboring the constructed plasmids. As predicted, both the constitutively overexpressed *cas9* gene and the induced gene led to significant growth retardation in bacterial cells (Figure 2B). After 48 h of cultivation, the strains without a functional Cas9 protein product, namely BLG100, BLG107, BLG108, and BLG109, reached a biomass of approximately 25 (OD600) without xylose induction. However, those strains with an induced Cas9 protein product accumulated a biomass of no more than 10. The strain with a constitutively produced Cas9 protein (BLG110) accumulated a biomass of merely 5.8. These results were consistent with the specific growth rates of the strains. As shown in Table 3, under uninduced conditions, the maximum specific growth rates of BLG107, BLG108, and BLG109 (1.32–1.40 h^−1^) were more than double those achieved under the induction conditions (0.49–0.60 h^−1^). In addition, the results acquired under the two conditions (uninduced and induced) were close to those for BLG100 and BLG110, respectively. This observation further indicated that the expression of the *cas9* gene placed a tremendous burden on the growth of bacterial cells, and the inactivation of *cas9* transcription contributed to the high transformation rates of the genome-editing plasmids, which resulted in enhanced propagation during the culture stage after transformation.

### 3.2. Efficient Genome Editing Was Triggered by 0.5% Xylose

The xylose-induced genome editing process essentially consisted of three steps. First, the obtained strains BLG107, BLG108, and BLG109 were cultivated in medium with 10 μg/mL erythromycin at 30 °C. All the strains reached a biomass of 10 in approximately 8 h, and, during this time, the genome-editing vector was able to massively replicate. Then, 0.5% xylose was added to the cultures of BLG107, BLG108, and BLG109 for Cas9 production, and the cultures were kept at 30 °C for an additional 24 h. It can be clearly observed that all the bacterial cells exhibited stunted growth, as the biomass hardly increased in this stage. In the second step, a pellet of the 24-h culture of the three strains was inoculated into fresh medium with 0.5% xylose but without antibiotics. These cultures were also maintained at 30 °C for 24 h, which, thereby, afforded sufficient time for recombination. Lastly, the cells were recovered and plated onto TB medium supplemented with 10 μg/mL erythromycin for selecting positive transformants.

Since functional Cas9 caused DSBs at target chromosomal sites, cells could only survive when the donor fragments were integrated into the DSB site-flanking locations. The typical feature of this intended genome-editing genotype is the coexistence of target cleavage and homologous repair. This can be verified by diagnostic PCR. Thus, chromosomal regions upstream and downstream of the designed homologous arms (not contained in the plasmid pN-xylamyLD100-102) were amplified. The native *amyL* gene on the chromosome generated a PCR product of 1901 bp, while an edited gene generated a product of 1417 bp (Figure 3A). The diagnostic PCR results showed that the efficiencies of target gene disruption at PAM621, PAM832, and PAM984 were 64.3%, 70.9%, and 47.1%, respectively (Figure 3B). These results suggested a sequence preference of the CRISPR/Cas9 system, which was potentially related to the PAM sequences and the immediately adjacent nucleotides, according to previous studies [24,25]. DNA sequencing of the homologous regions of the transformants further indicated that the genome-editing system functioned efficiently.

The transformants with expected editing efficiency lost the genome-editing plasmids at elevated cultivation temperatures. After 24 h of incubation at 42 °C, nearly all of the isolated colonies (94.8%) were sensitive to 10 μg/mL erythromycin, which indicated loss of the plasmids. Plasmid loss was verified by colony PCR, and the bands generated by all the examined colonies were the same as those generated in the previous diagnostic PCR. One colony was randomly selected and sent for sequencing. As predicted, a 484-bp fragment was removed from the *amyL* ORF (Appendix A). We also transferred pN-sgRNA1, which contained only an *amyL*-targeting sgRNA module into *B. licheniformis*. No colonies could be obtained on the selective agar plate, which indicates that the plasmid was unable to generate the *amyL* deletion mutant without a repaired DNA template for homologous recombination.

### 3.3. Low Temperatures Increased the Genome-Editing Efficiency

*B. licheniformis* has an optimal growth temperature of 50 °C, which is higher than those of *E. coli* and *Saccharomyces cerevisiae*. In previously constructed CRISPR systems, the working temperature was chosen according to the optimal growth temperature of the cell host, which ensures a certain success rate for genome editing. On the other hand, recent studies suggested that temperature, in addition to growth rate, could regulate CRISPR-Cas activity in the cell host [26,27,28]. To assess whether the optimal growth temperature of *B. licheniformis* is beneficial for the performance of the CRISPR system, we tested inducible genome editing at 20 °C, 25 °C, 30 °C, 33 °C, 37 °C, 42 °C, and 50 °C using BLG107. Considering that the expression of the *cas9* gene and genome editing occurred simultaneously at specific temperatures and that the thermosensitive replication of the vector was also sensitive to temperature, we separated the whole process into different stages and investigated the effect of temperature on gene expression, Cas9 protein activity, and thermosensitive replication of the plasmid (Table 4).

In the first round of cultivation, the pre-cultivated cells (at 30 °C for 10 h without xylose) were equally divided into seven aliquots and cultured at the previously mentioned different temperatures with the addition of 0.5% xylose. Since the cultures exhibited similar biomasses (~5) at the end of cultivation (24 h), they were serially diluted 10-fold to 10-6 and plated onto TB agar. Different numbers of single colonies were formed on the plates on the second day. These colonies were replica plated on TB selective agar (with 10 μg/mL erythromycin) to test their sensitivity to antibiotics. Transformants obtained on the selective agar were regarded as target gene-edited candidates and were subjected to diagnostic PCR for verification. It is clear that temperature has a significant influence on the expression of the *cas9* gene. The highest level of expression was detected at 33 °C, which was over three-fold that at 20 °C, and then, the expression level exhibited a sharp decrease at 37 °C or higher. However, in our previous study, the xylose-inducible promoter exhibited stronger expression at temperatures higher than 37 °C in *B. licheniformis*. This inconsistency could be explained by severe loss of the thermosensitive plasmid harboring a xylose-inducible expression box. The results of colony sensitivity to erythromycin clearly indicated that most bacterial cells had lost the plasmid at 37 °C or higher. Cells harvested and subjected to RT-qPCR detection under these conditions harbored little or no plasmid, which led to extremely low relative expression of the *cas9* gene. When we performed diagnostic PCR to verify the genotype of the colonies obtained on TB agar, only a small proportion of the colonies were seen to be successfully edited at the intended genomic loci.

In the second round of cultivation, a pellet of the 24-h culture was inoculated into fresh medium with 0.5% xylose. These cultures were maintained at different temperatures for another 24 h. The temperature showed a similar influence on *cas9* gene expression and plasmid maintenance. However, the successful ratio of targeted editing was sharply improved, especially at lower temperatures. Furthermore, 33 out of 34 colonies obtained from TB agar at 20 °C exhibited the intended double-crossover genotype, which indicated that all the processes, including *cas9* gene expression, Cas9-guided DSB, and homologous repair, were sufficiently completed. Nevertheless, the constructed CRISPR system was hardly valid at high temperatures in *B. licheniformis*. A temperature of 50 °C led to complete failure of genome editing, which indicated extreme difficulties in the functioning of the system.

In summary, the above results proved that temperature was a pivotal factor in the functioning of the constructed CRISPR system. Although the expression of the *cas9* gene was relatively weak at lower temperatures, the accumulated Cas9 protein was sufficient for adaptation to the target gene and generation of DSBs. Equally importantly, a second round of cultivation at a lower temperature was beneficial for homologous recombination repair. This is consistent with a recent finding that a lower temperature could buy a bacterium time for successful functioning of its CRISPR-Cas machinery [28].

### 3.4. The Genome-Edited Strain Exhibited the Predicted Phenotype

The effect of *amyL* disruption in the genome on cell growth and amylase production was investigated by inoculating the parent and randomly selected mutant strains into fermentation medium and detecting the biomass and amylase activity in the supernatant at set intervals. As shown in Figure 4, both the parent and the edited strain entered the logarithmic phase after 2 h. These strains attained maximal biomasses of 9.3 and 9.2, respectively. The strains maintained almost the same biomass during the stationary phase. The only difference was that the edited strain reached its maximal biomass 2 h later (18 h) than the parent strain (16 h). These results suggested that the disruption of *amyl* had a marginal effect on the growth of the bacterial cells. Amylase-encoding genes from various origins have been widely subjected to deletion mutation for either analysis of gene function or optimization of microbial cell factories [29,30,31]. All published results indicate amylase-encoding genes as being dispensable for cell growth, especially in the case of *Bacillus* strains, which usually have complex carbohydrate hydrolysis enzyme systems. The results also indicated that the genome-editing system developed in this study did not negatively impact cell growth.

As expected, the extracellular amylase activity decreased dramatically after genome editing. Less than 3% of the initial activity remained in the mutant strain. We further characterized the residual extracellular enzymes, and the hydrolysis product analysis showed that these enzymes were mainly maltogenic-type amylases (data not shown) and, thus, could not be the product of *amyL*. The α-amylase encoded by *amyL* is the main secreted protein in *B. licheniformis*. In addition, this species also produces other hydrolytic enzymes, which degrade starch by cleaving maltose units preferentially [32,33]. The features of this enzyme are highly consistent with those of the residual enzymes detected in this study. On the other hand, *B. licheniformis* produces many different types of secreted hydrolytic enzymes to efficiently utilize macromolecular substrates in the environment. Although this trait beneficially contributes to the industrial applications of this strain, it is inconvenient for product extraction when the strain is used in enzyme production.

## 4. Discussion

In recent years, the CRISPR-Cas9 system has been developed as a powerful tool for high-efficiency genome editing [34,35]. Two critical components must be introduced to apply the system, which include the Cas9 protein and gRNA. Therefore, the appropriate choice of a delivery system for the above two components is critical for genome-editing systems to function effectively in targeted cells. Although multiple delivery approaches, including viral delivery, physical delivery, chemical delivery, and encapsulation, have been developed for different species, the most common method for microorganisms remains the delivery of genetic material that instructs the target cell to produce its own Cas9/sgRNA ribonucleoprotein complex in vivo [36]. At this time, there are problems regarding this production process, as the expression of heterologous genes could inevitably affect cell growth and proliferation. The template-driven nature of gene expression also limits the control of the total amount of Cas9 protein. Furthermore, sgRNA, and off-target cleavage, which in some studies was greater than 50%, has been attributed to excess dosing [37,38,39]. Inducible CRISPR systems, on the other hand, which allow conditional gene expression, introduce great flexibility to the standard CRISPR strategy. We have developed an all-in-one plasmid-based CRISPR system in *B. licheniformis* with the *cas9* gene under the control of a xylose-inducible promoter that allows tight spatiotemporal control of genome modification using an oligo sugar input. The plasmids effectively replicate in the first step without inducer and add no burden on cell growth. With limited culture volume (25-mL culture in a 250-mL flask) in a shaking flask equipped with baffles, the specific growth rate of strains harboring the all-in-one plasmids was comparable with that of the wild-type strain (>1 h^−1^), which greatly improved the efficiency in transformation. After *B. licheniformis* was successfully transformed with the plasmid carrying a *cas9* gene, the 1368-amino acid protein could be rapidly produced in the presence of xylose. Since xylose is able to induce the expression level at a range of 1–200 folds [40], this design has provided better flexibility in modulating genome editing. This system could also be especially useful for editing genes that are essential in the early stage of growth.

One of the perpetual concerns of CRISPR technology is editing specificity. In the context of in vivo editing specificity, the first concern is genome-targeting accuracy. The Cas9 protein is assumed to act only on the gene assigned by the gRNA when it accumulates in the host cell. It has been reported that a plasmid-delivered CRISPR/Cas9 system would generate more off-target effects than mRNA-delivered or protein-delivered systems [41,42]. The specificity could be improved through various strategies such as using truncated guide sequences [43], paired double nicking [44], and Cas9 modification [45,46]. One common method is the restriction of the the mature DNA endonuclease dosage to meet the demand of editing at the intended locus rather than acting on excess off-target sites. In this case, we assessed the feasibility of such an approach by applying a xylose operon. The introduced pN100 plasmid, with only an *amyL*-targeting sgRNA module but no donor DNA template, generated no transformants. The NHEJ system actively rescues cells with off-target DSBs during the stationary phase. This is the time at which Cas9 is produced under xylose induction, which indicates that the regulated expression of the *cas9* gene greatly aids the specificity of genome editing.

Temperature acts as an environmental regulator of the CRISPR/Cas9 system. Previous results suggested that the activity of an introduced constitutive CRISPR-Cas system is directly related to cell growth. When sgRNA and Cas9 protein are prepared in vitro and transfected into cells to do genome editing, temperature mainly influence the editing efficiency by modulating Cas9 activity. The results showed that higher temperatures were better [27]. On the contrary, when Cas9 proteins need to be produced in vivo, it seems that that lower temperatures would improve the performance of this system [28]. In this case, a reduced growth rate caused by a low temperature promotes the accumulation of CRISPR-Cas complexes, which ensures high genome editing activity. However, the mechanism by which temperature modulates the conditional CRISPR-Cas system has not been studied. We previously showed that elevated temperatures (>37 °C) resulted in elevated transcription of the xylose operon in *B. licheniformis* [47]. The apparent thermophilic characteristics of both cell growth and the promoter raise the question of whether the conditional CRISPR-Cas system also benefits from high temperature. We found that relatively high CRISPR-Cas-mediated genome editing efficiencies occurred at relatively low temperatures (<30 °C). Because the temperature has direct effects on multiple fundamental biological processes, we observed that there was, in general, a compromise between the production and function of the Cas9 protein in vivo. By investigating the gene expression intensity, plasmid loss, and the target gene editing rate at different temperatures, we found that a lower temperature could buy a bacterium more time for successful functioning of the CRISPR-Cas machinery.

In conclusion, a conditional CRISPR/Cas9 system was constructed in *B. licheniformis*. The 4107 bp-*cas9* gene was incorporated into a thermosensitive shuttle vector and was under the strict control of a xylose-inducible promoter. This design has two advantages over the commonly used constitutive expression pattern in the CRISPR/Cas9 system. First, a silent *cas9* gene without induction improves the success rate when *B. licheniformis* is transformed with the all-in-one plasmid of a large size. Second, efficient genome editing can be triggered by the inducer xylose, which increases flexibility in the control of DNA nuclease dosage.

## Figures and Tables

**Figure 1 microorganisms-08-00754-f001:**
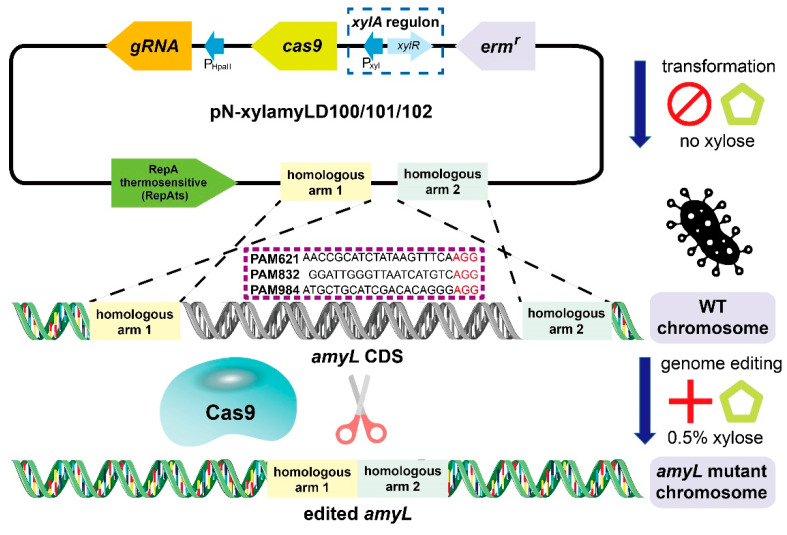
Deletion of gene fragments from the *B. licheniformis* chromosome using a controllable clustered regularly interspaced short palindromic repeat (CRISPR)/Cas9 system. In the genome-editing plasmid, the expression of the *cas9* gene is controlled by a *xylA* regulon, including the xylose-inducible promoter P_xyl_ and repressor-encoding gene *xylR*. The sgRNA is inserted downstream of the gram-positive promoter P_HpaII_. The customized CRISPR/Cas9 vectors target the *amyL* gene at 3 protospacer-adjacent motif (PAM) loci. Gene fragment deletion was achieved via a homologous recombination event. Two homologous arms connected by black dashed lines show the approximate regions of recombination (∼300 bp each), and they were incorporated into the CRISPR/Cas9 vector to help cells survive chromosomal double-strand breaks (DSBs). The deleted region within the *amyL* CDS is shown in grayscale. P_HpaII_: the constitutive promoter from pMA5. P_xyl_: the xylose-inducible promoter from *B. licheniformis*. *cas9*: the *S. pyogenes cas9* gene amplified from the plasmid pCas9. gRNA: the chimeric gRNA with a sequence targeting the open reading frame (ORF) of the *amyL* gene. *erm^r^*: the erythromycin resistance marker gene.

**Figure 2 microorganisms-08-00754-f002:**
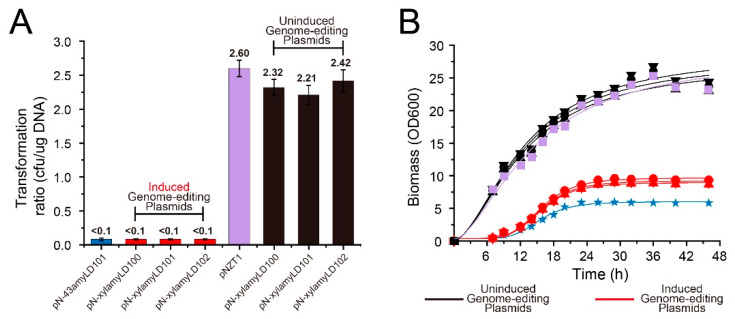
Transformation ratio of the different constructs (**A**). The genome-editing plasmids pN-xylamyLD100, pN-xylamyLD101, and pN-xylamyLD102 were used to transform *B. licheniformis* under induced and uninduced conditions. pN-43amyLD101 and pNZT1 were used as the control sets. Transformation efficiency was calculated as the number of transformants per µg of plasmid DNA. Growth of the recombinant strains (**B**). ■ BLG100, ● BLG109-uninduced, ▼ BLG108-uninduced, ▲ BLG107-uninduced, ★ BLG110, ● BLG109-induced, ▼ BLG108-induced, and ▲ BLG107-induced. BLG100, BLG110, BLG107-uninduced, BLG108-uninduced, and BLG109-uninduced cells were cultured in standard medium, while BLG107-induced, BLG108-induced, and BLG109-induced cells were cultured in medium containing 0.5% xylose. Then, 250-mL shaking flasks with baffles containing 25 mL of medium were inoculated with specific strains. Biomass was monitored by measuring the absorbance at 600 nm (OD600). The cell culture samples were diluted to suitable concentrations before detection to maintain the absorption reading range of the spectrophotometer between 0.2 and 1.2. The results are expressed as the mean of three replicates (*n* = 3).

**Figure 3 microorganisms-08-00754-f003:**
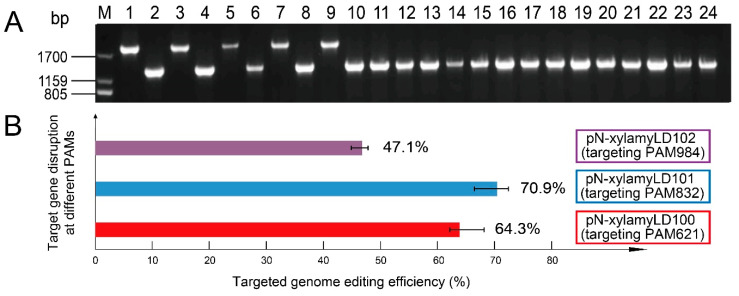
Confirmation of genome editing. Diagnostic PCR of the transformants. PCR products using randomly selected colonies as the templates were labeled as 1-24. The edited *amyL* gene generated smaller band sizes than the native gene. The sizes of the DNA markers are labeled on the left (**A**). The efficiencies of target gene disruption at PAM621, PAM832, and PAM984, which were calculated as the number of successfully edited colonies divided by the number of colonies obtained in the selection plates (**B**).

**Figure 4 microorganisms-08-00754-f004:**
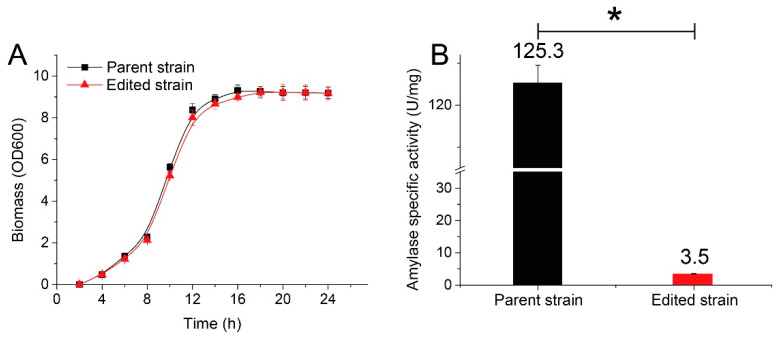
Growth curves of the parent *B. licheniformis* and the edited strain with a disrupted *amyL* gene (**A**) and their extracellular amylase activities after fermentation (**B**). Luria-Bertani (LB) broth was inoculated with 1% of the culture and incubated at 37 °C with shaking at 250 rpm. Bars with * represent statistically significant differences between groups at *p* < 0.01 (*n* = 3).

**Table 1 microorganisms-08-00754-t001:** Bacterial strains and plasmids used in this study.

Strain or Plasmid	Description ^1^	Source/Reference ^2, 3^
Strains		
*Escherichia coli* JM109	*F*′, *traD36, proAB +. lacIq, Δ (lacZ), M15/Δ (lac-proAB), glnV44, e14−, gyrA96, recA1, relA1, endA1, thi, hsdR17*	CICIM-CU
*Bacillus licheniformis* DSM13	Wild-type	CICIM-CU
BLG100	*B. licheniformis* DSM13 harboring pNZT1	This work
BLG107	*B. licheniformis* DSM13 harboring pN-xylamyLD100	This work
BLG108	*B. licheniformis* DSM13 harboring pN-xylamyLD101	This work
BLG109	*B. licheniformis* DSM13 harboring pN-xylamyLD102	This work
BLG110	*B. licheniformis* DSM13 harboring pN-43amyLD100	This work
Plasmids		
pMD19-T	*E. coli* cloning vector, Ap^R^, 2692 bp	TaKaRa
pCas9	*E. coli* vector with a minimal CRISPR3-Cas9, the leader sequence, and a repeat-spacer region, 12671 bp	[21]
pE194	Replication-thermosensitive *Bacillus* vector, 3728 bp	[22]
pNZT1	pE194-derivative, *Bacillus* vector, Em^R^, 4069 bp	[22]
pMA5	*E. coli/Bacillus* shuttle vector, Neo^R^/Ap^R^, *P_HpaII_*, 7107 bp	BGSC
pHY300-PLK	*E. coli*/*Bacillus* shuttle vector, Ap^R^/Tet^R^, 4870 bp	CICIM-CU
pN-sgRNA1	pNZT1 derivative with an sgRNA targeting the PAM621 locus, 4550 bp	This work
pN-sgRNA2	pNZT1 derivative with an sgRNA targeting the PAM832 locus, 4549 bp	This work
pN-sgRNA3	pNZT1 derivative with an sgRNA targeting the PAM984 locus, 4549 bp	This work
pM-xylACas9	pMD18 derivative with the xylose regulon and *cas9*, Ap^R^, 8232 bp	This work
pN-sgRNA1-xylACas9	pNZT1 derivative with an sgRNA targeting the PAM621 locus and *cas9* under the control of the xylose regulon, 10090 bp	This work
pN-sgRNA2-xylACas9	pNZT1 derivative with an sgRNA targeting the PAM832 locus and *cas9* under the control of the xylose regulon, 10089 bp	This work
pN-sgRNA3-xylACas9	pNZT1 derivative with an sgRNA targeting the PAM984 locus and *cas9* under the control of the xylose regulon, 10089 bp	This work
pN-xylamyLD100	pNZT1-derivative genome-editing plasmid targeting the PAM621 locus of amyl with *cas9* under the control of the xylose regulon, 10880 bp	This work
pN-xylamyLD101	pNZT1-derivative genome-editing plasmid targeting the PAM832 locus of amyl with *cas9* under the control of the xylose regulon, 10879 bp	This work
pN-xylamyLD102	pNZT1-derivative genome-editing plasmid targeting the PAM984 locus of amyl with *cas9* under the control of the xylose regulon, 10879 bp	This work
pN-43amyLD100	pNZT1-derivative genome-editing plasmid targeting the PAM621 locus of amyl with *cas9* under the control of the P43 promoter, 9758 bp	Our Lab

^1^ Ap^R^, ampicillin resistance. Tet^R^, tetracycline resistance. Neo^R^, neomycin resistance. Em^R^, erythromycin resistance. ^2^ BGSC, *Bacillus* Genetic Stock Center ^3^ CICIM-CU, Culture and Information Center of Industrial Microorganisms of China Universities (http://cicim-cu.jiangnan.edu.cn/).

**Table 2 microorganisms-08-00754-t002:** Primers used in this study ^1^.

Primer	Sequence (5′-3′)
pNZT-F	tacctatcacctcaaatggttcgc
pNZT-R	ctctagaggatcccaccgcg
sgRNA-F	cgcggtgggatcctctagagttttgagtgatcttctcaaaaaatactacc
sgRNA-R	accatttgaggtgataggtaaaaaaaagcaccgactcggtg
xylA-F	tcctctagagatatcgtcgaccgttttgggattttatcaacaatc
xylA-R	cttatccattccgatctcccccttcactt
Cas9-F	gggagatcggaatggataagaaatactcaataggcttaga
Cas9-R	cttgcatgcctgcaggtcgactcagtcacctcctagctgactcaa
HA1-F	attctgcagcagcggcggca
HA1-R	gcttaaaccatgtttggacagaattgatgacaaccggctc
HA2-F	gagccggttgtcatcaattctgtccaaacatggtttaagc
HA2-R	gaattgatgacaaccggctcc
HASOE-F	atactcgagattctgcagcagcggcggca
HASOE-R	atcctcgaggaattgatgacaaccggctc
diag-F	atgaaacaacaaaaacggct
diag-R	ctatctttgaacataaatt

^1^ F, upstream primer; R, downstream primer.

**Table 3 microorganisms-08-00754-t003:** Maximum specific growth rates of the recombinant strains.

Strains	Maximum Specific Growth Rates (h^−1^) ^1^
BLG110	0.54 ± 0.02
BLG107-induced	0.49 ± 0.03
BLG108-induced	0.49 ± 0.01
BLG109-induced	0.60 ± 0.02
BLG107-uninduced	1.32 ± 0.05
BLG108-uninduced	1.40 ± 0.04
BLG109-uninduced	1.36 ± 0.06
BLG100	1.21 ± 0.03

^1^ BLG100, BLG110, BLG107-uninduced, BLG108-uninduced, and BLG109-uninduced cells were cultured in standard medium, while BLG107-induced, BLG108-induced, and BLG109-induced cells were cultured in medium containing 0.5% xylose. Biomass was monitored by measuring the absorbance at 600 nm (OD600) at 0, 7 h, 9 h, 12 h, 14 h, 16 h, 18 h, 20 h, 23, 26 h, 29 h, 32 h, 36 h, 40 h, and 46 h. The cell culture samples were diluted to suitable concentrations before detection to maintain the absorption reading range of the spectrophotometer between 0.2 and 1.2. The growth curves were fitted by a logistic growth model using Origin 9.0, and μ was calculated by the equation μ = ∆lnOD600/∆*t*, where *t* is the time. The results are expressed as the mean of three replicates (*n* = 3).

**Table 4 microorganisms-08-00754-t004:** Effect of temperature on gene expression, Cas9 protein activity, and thermosensitive replication of the plasmid in the conditional clustered regularly interspaced short palindromic repeat (CRISPR)/Cas9 system-guided genome editing process ^1^.

Cultivations	T (°C)	Relative Expression of the *cas9* Gene	Colonies on TB Agar	Colonies on Selective TB Agar	Target Gene Edited
**ROUND 1** **0.5% xylose for 24 h**	20	5.12 ± 0.11	30 ± 1	30 ± 1	2 ± 1
	25	8.63 ± 0.35	31 ± 2	32 ± 1	1 ± 1
	30	15.92 ± 0.42	33 ± 2	33 ± 2	3 ± 1
	33	16.26 ± 0.55	34 ± 1	27 ± 1	1 ± 1
	37	7.9 1± 0.28	33 ± 1	4 ± 1	0 ± 1
	42	3.37 ± 0.12	32 ± 1	0	0
	50	0	30 ± 1	0	0
**ROUND 2** **0.5% xylose for 24 h**	20	3.32 ± 0.09	34 ± 1	34 ± 1	33 ± 1
	25	5.57 ± 0.22	37 ± 1	37 ± 2	30 ± 1
	30	10.13 ± 0.52	39 ± 2	39 ± 2	26 ± 1
	33	10.91 ± 0.37	44 ± 2	44 ± 2	27 ± 1
	37	5.32 ± 0.20	41 ± 1	2 ± 1	12 ± 1
	42	0	38 ± 1	0	1
	50	0	38 ± 1	0	0

^1^ The results are expressed as the mean of three replicates (*n* = 3).

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
