# Peer review of "Efficient Genome Editing in Bacillus licheniformis Mediated by a Conditional CRISPR/Cas9 System"

_microorganisms, 2020, doi:10.3390/microorganisms8050754_

Round 1
Reviewer 1 Report
This paper Efficient Genome Editing in Bacillus licheniformis Mediated by a Conditional CRISPR/Cas9 System describes the single-step plasmid transformation of B. licheniformis cells using CRISPR/Cas9 system. The topic of the study is interesting and the experiments were well designed, prepared and described. However, some minor issues require improvements before publication in Microorganisms. For this purpose see the comments below.
General comments:
Please pay attention to Latin phrases (B. licheniformis, in vivo, etc.) to write it in italic in the whole manuscript.
Introduction:
In my opinion, the Introduction is too long and should be shortened to one manuscript page (around 50 lines of text instead).
Materials and methods:
A more detailed description of biomass measurements is missing. The authors just mention about OD600 measurements in Section 2.5, while the results of OD600 measurements are presented in sections 3.2 and 3.5.
Results:
Section 3.1: should be moved to Materials and methods (e. g. section 2.4), while this section does not present any results.
Section 3.2: is a bit confusing. The results described as ‘next’ (line 214) are presented as first at the graph, and that might be misleading, due to similar names of plasmids used.
Line 238 and Fig. 2B: As I mentioned, those measurements are not well described in Materials and methods section. My biggest concern is the values of obtained Biomass (OD600). What do you mean by biomass? Is this value of absorbance at OD600? If yes, please keep in mind that such high values of absorbance are unreliable for most of the apparatus, due to the non-linearity of many instruments over absorbance higher than 2.0. Please add an explanation to that in the manuscript.
Table 3: How did you calculate the maximum specific growth rate? What do you mean by this unit? Is that generation per hour (should be explained at least in the table footnote). Moreover, most of the literature indicates generation time during the expotential phase for this strain equal to 2-3 hours. Please explain and discuss in the manuscript why in your experiment this time is almost 3 times lower for uninduced strains (around 40 min).
Figure 3B: Please correct the axis description on this graph as there is something wrong ( no titles and x-axis only to50?).
Please pay also attention to missing references for some of your statements e. g. page 5, line 148; page 12, line 370, or missing words like page 10, line 302: ‘above (mentioned) different temperatures…’
The authors should carefully verify the References list (e.g. reference 8).
Reviewer 2 Report
The manuscript by Li et al. “Efficient Genome Editing in Bacillus licheniformis Mediated by a Conditional CRISPR/Cas9 System” describes the development of a CRISPR/Cas9 genetic tool suitable for effective genome manipulation in the aforesaid organism. The authors developed constructs for the knockout of the amylase gene in which the Cas9 gene is placed under the xylose-inducible promoter. The authors demonstrate that the repression of Cas9 is crucial for effective transformation of B. licheniformis cells, which is, of course, a prerequisite to efficient genome engineering. For the most part, this work is based on sound experiments. However, I am skeptical about the conclusions drawn from the experiments showing the “dependence” of CRISPR/Cas9 system on the temperature. Notably, the plasmid from which the system is expressed contains a TS origin of replication and should be cured from the cells at elevated temperatures. The plasmid loss should be apparent from the biomass after the experiment because the toxic effect of Cas9 should be absent in the cells cured of the plasmid (the authors do make a remark on the growth rate, line 306, correlating the growth rate with the temperature; however, the Cas9 expression also affects the growth rate and these two variables are inseparable in the experiment such that it is). Another unexplored possibility is that the plasmid keeps on accumulating mutations because the elevated temperature becomes the selecting agent. It would be important to determine definitively if the lowest temperature (20 C) is truly the optimal one for CRISPR/Cas9 editing, because this result is in opposition to earlier results reported in Xiang et al 2017 (ref 36). There, the authors have determined that, unlike ZFN and TALEN, CRISPR-Cas9 editing efficiency is significantly hindered by low-temperature treatment.
Other major remarks:
The manuscript would benefit from thorough re-writing. Abstract and Introduction sections are poorly structured with unusual statements that are explained only retroactively.
Specific remarks:
Line 16-17 “produces native enzymes or metabolites” – is used for the production of?
Lines 17-19 “An extremely low transformation efficiency undermines successful genetic manipulation in B. licheniformis, even if the same method is easily applied in other genera when the genetic background of the source host is well known – This is a very confusing sentence which is coincidentally repeated in the Introduction section. Please consider rephrasing.
Lines 20-21 “The transformation ratio was improved …with the conditional genome-editing plasmid” – at this point the reader is not aware that the authors have developed a genome editing tool in which the Cas9 expression can be repressed/induced. Please consider restructuring. Also, a “genome editing” plasmid may be too colloquial.
Lines 24-25 “Successful genome editing was triggered by 0.5% xylose” – similar to earlier comment. The reader is unaware of the inducible promoter.
Lines 24-25 “… and the efficiencies of amyL gene disruption targeting 3 PAMs were 64.3%, 70.9% and 47.1%. – I believe amyL and “PAM” should be defined.
Lines 25-27 – “Genome editing was regulated by the temperature-related growth rate. The maximum success rate reached 94.4% at 20°C” – from the experimental point, I am unsure about this interpretation. In the context of the abstract, it will be unclear to the reader what the authors have in mind.
Line 37 - together contribute nearly half of the – to is missing. Contribute together to nearly half..
Line 44 – "in only" – Order. Only in the protoplasm..
Line 50 – What is “rare competence development”?- please clarify.
Lines 71-72 “This characteristic of precise genome editing makes it an ideal tool for genome engineering” – I’m unsure that the authors presented a sufficient number of argument to reach this point.
Lines 75-76 “CRISPR/Cas9 system has been widely applied for either gene deletion or insertion in bacteria such as E. coli and B. subtilis” – Is CRISPR/Cas9 really comparable to how much work has been and is still being done with the Wanner Datsenko method in E. coli?
Line 83 – probably a word missing: success of the transmembrane transport?
Line 93- I find the construct “Bacillus-inducible” very unusual.
Line 288 – Figure 3 legend: Confirmation, not “conformation”. Please clarify what the numbers above the gel stand for.
Round 2
Reviewer 2 Report
The manuscript is significantly improved.